# Wave Energy Assessment in the South Aquitaine Nearshore Zone from a 44-Year Hindcast

**Ximun Lastiri** [1] , **Stéphane Abadie** [1,*], **Philippe Maron** [1], **Matthias Delpey** [2], **Pedro Liria** [3], **Julien Mader** [3] **and Volker Roeber** [1]

1   SIAME Laboratory, E2S-UPPA-HPC Wave Chair, Université de Pau et des Pays de l'Adour,
    64600 Anglet, France; ximun.lastiri@univ-pau.fr (X.L.); philippe.maron@univ-pau.fr (P.M.);
    volker.roeber@univ-pau.fr (V.R.)
2   Center Rivages Pro Tech, SUEZ Eau France, 64210 Bidart, France; matthias.delpey@suez.com
3   Azti-Tecnalia, Marine research division, 20110 Pasaia, Spain; pliria@azti.es (P.L.); jmader@azti.es (J.M.)
*   Correspondence: stephane.abadie@univ-pau.fr

**Abstract:** Wave resource assessment is the first step toward the installation of a wave energy converter (WEC). To support initiatives for wave energy development in the southwest of France, a coastal wave database is built from a 44-year hindcast simulation with the spectral wave model SWAN (Simulating WAve Nearshore) run on a high-resolution unstructured grid. The simulation includes shallow-water processes such as refraction, shoaling, and breaking. The model is validated against a five-year coastal wave buoy recording. The study shows that most of the resource is provided by sea states with wave heights ranging from 2 to 5 m, with wave periods from 10 and 15 s, and coming from a very narrow angular sector. The long hindcast duration and the refined unstructured grid used for the simulation allow assessment of the spatiotemporal distribution of wave energy across the coastal area. On the one hand, large longshore variations of the resource caused by steep bathymetric gradients such as the Capbreton submarine canyon are underlined. On the other hand, the study highlights that no specific long-term trend can be extracted regarding the coastal wave energy resource evolution. The provided downscaled local wave resource information may be used to optimize the location and design of a future WEC that could be deployed in the region.

**Keywords:** wave energy; numerical model; coastal propagation; SWAN

---

## 1. Introduction

To meet the objectives that policymakers set regarding the reduction of human carbon footprint, the development of new energy sources is mandatory. Wave energy conversion is considered as one of the possible marine renewable energy contributions to a sustainable energy mix in the future. France has a very long coastline of about 2400 km, which was shown to offer a significant wave energy resource [1]. The wave energy resource is quite well distributed along the shoreline as opposed to, e.g., tidal energy where only a few hotspots are suitable for installation. Additionally, available wave energy during the year is usually the highest when the energy demand is also high, i.e., in winter.

The first step toward wave energy exploitation is the assessment of the wave energy resource on site. Wave resources were already quite well studied at a regional scale for the northern part of Spain, from Galicia to Basque Country [2–4], as well as locally along the western coast of France, e.g., in Le Croisic area [5]. When assessing the available resource, its space–time variability is a key parameter. This variability may be especially strong in the coastal area due to wave–bottom interactions, which may induce large variations in the wave field both in space and in time. Space variability is obviously interesting for the installation of a wave energy converter (referred to as WEC in the rest of this paper)

as it is of the utmost importance to determine an optimal WEC location [6]. The time variability is also crucial to properly anticipate the energy effectively available throughout the year, as major seasonal differences may be observed in some regions [7]. Finally, the extreme sea states and long-term variability of the wave climate should also be addressed to ensure that the sizing of the WEC is compatible with survivability in both current and future conditions [8].

The objective of the present study is to provide an assessment of the local wave energy resource distribution along the south Aquitaine coastal area, a region that was identified as potentially suitable for WEC deployment by previous studies [9]. To that end, a numerical modeling strategy is deployed using a fine computational domain based on an unstructured mesh to capture the different processes affecting the wave field over the shelf, especially the effect of a remarkable topographic feature: the Capbreton submarine canyon. The 44-year duration of the obtained high-resolution database is also used to analyze long-term trends of the local wave resource.

The paper is organized as follows: Section 2 describes the data used, the parametrization of the model, and its validation. The analysis of the obtained results is detailed in Section 3. Those results are discussed in Section 4, and conclusions are drawn in Section 5.

## 2. Materials and Methods

### 2.1. Offshore Wave Data

The present study relies on a new wave hindcast dataset produced with a spectral wave model to provide a refined long-term database of coastal sea states in the studied region. The data used to prescribe offshore boundary conditions in our wave hindcast simulation were from the so-called BOBWA (Bay Of Biscay Wave Atlas) dataset covering 44 years (i.e., from 1958 to 2001). The BOBWA database was constructed from a hindcast simulation with a North Atlantic wave modeling system based on the WAVEWATCH III®code and forced by 40-year European Center for Medium-Range Weather Forecasts Re-Analysis (ERA-40) wind fields, providing refined spatial resolution (0.5° for water depth superior to 4000 m and 0.1° elsewhere) over the Bay of Biscay [10]. The validation of the wave model was based on 10 buoys moored at different locations across the Bay of Biscay, at depths ranging from −17 m to −54 m. One deep-water buoy was also used (water depth −4500 m). The computed wave fields were then used in the analysis of mean and extreme wave height trends and variability. The BOBWA dataset provides hourly wave spectra at several points along the south Aquitaine coastal area studied in this paper, from Hendaye at the Spanish border up to Contis approximately 80 km north. In order to use this incident wave information properly, the offshore limit of our computational domain was adjusted to the most onshore BOBWA output points as illustrated in Figure 1. The offshore boundary of our coastal model included 10 of these points, thus allowing for a spatially varying boundary condition. As the next BOBWA output points are located much farther offshore, an alternative (more offshore) delimitation of our computational domain would require significantly increasing the model complexity by taking into account additional forcing such as wave generation by wind among others; at the same time, it would require lowering model resolution to limit computation time. Considering this, the preferred compromise here was to focus on the coastal area and the related specific processes.

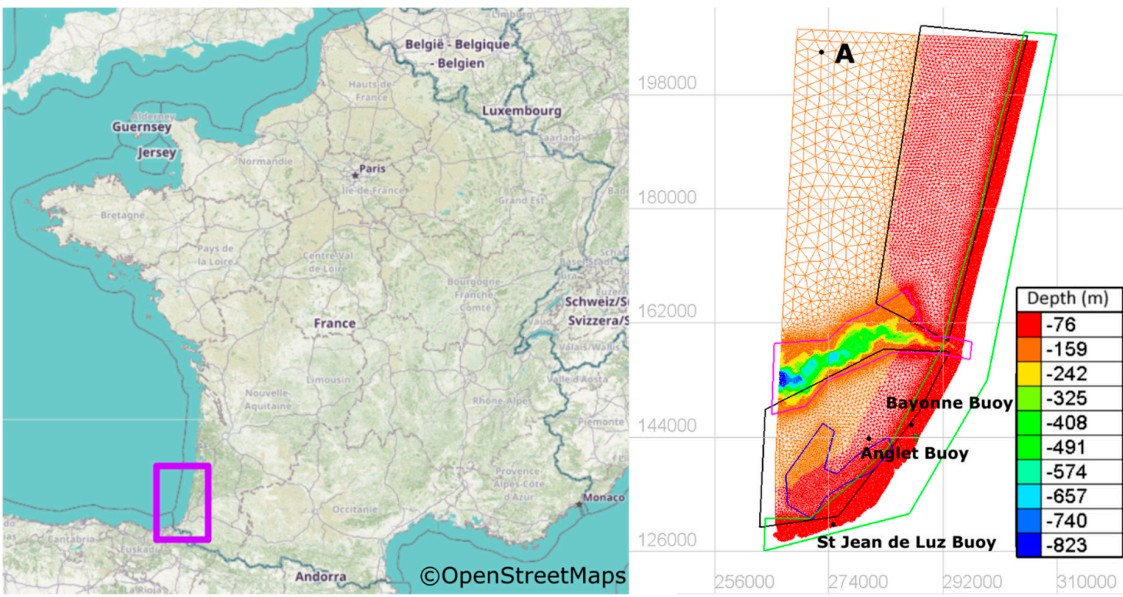

**Figure 1.** Unstructured mesh used in the present study, with bathymetry given by the color scale. Superimposed polygons indicate areas with different spatial resolution.

## 2.2. Validation Data

Measurements from a non-directional wave buoy moored off Bayonne port in 20-m depth were used for validation. The wave buoy provided almost 8000 wave observations from 1989 to 1994 (approximately evenly distributed). These data were used to verify the validity of our model set-up, especially its ability to properly represent wave propagation in limited water depth up to the inner shelf.

## 2.3. Computational Grid

The model used here was built with the code SWAN in its unstructured version [11,12]. SWAN (Simulating WAve Nearshore) is a state-of-the art third-generation wave propagation model dedicated to the study of coastal and nearshore regions. It is based on solving the conservation of the wave action density by means of a finite difference scheme. The numerical scheme used was the so-called cyclic scheme of Stelling and Leendertse, which reverts to BSBT (backward space/backward time) in the neighboring of open boundaries, land boundaries, and obstacles. To solve local bathymetric features that may affect the wave field during coastal propagation, an optimized unstructured mesh was used for the computation. Figure 1 shows the mesh implemented on the study area with the bathymetry indicated by the color scale. In this figure, the Capbreton submarine canyon is the most obvious feature in terms of depth variation, cutting the studied domain in two with a roughly east–west orientation. The canyon is characterized by strong bathymetric gradients, with depth dropping from about 100 m up to 800 m in the middle of the shelf. The coast is composed of sandy beaches north of Bayonne (Figure 1) usually associated with little bathymetric variations in the longshore direction, while rocky coasts are found down in the south where the bathymetry is more heterogeneous in both longshore and cross-shore directions. The bathymetry was assumed to be constant over the hindcast period.

The bathymetry initial dataset was composed of several inputs unevenly resolved, provided by different surveys including SHOM (French Naval Hydrographic and Oceanographic Service) data and several additional surveys with locally higher resolutions. The unstructured mesh featured 45,000 nodes distributed irregularly in space according to the different areas presented in Figure 1. These areas were determined by inspecting the depth gradient; the domain was divided into several areas based on averaged depth gradient characteristics and then using specific adapted constant mesh values in each area.

More precisely, north of the canyon, the grid step was set to 150 m from the shore to −20 m depth, 600 m from −20 m to 80 m, and 2000 m beyond. South of the canyon, the grid step was 150 m from shore to −40 m, 300 m in the blue area depicted in Figure 1 where the bathymetry has quite sharp variations, and 600 m beyond. Over the canyon, the grid step was set to 100 m to solve locally very strong bathymetric gradients. Finally, this mesh allowed resolving the main bathymetry features while keeping the computational time within reasonable limits and avoiding convergence problems sometimes met with high-energy events.

## 2.4. Parametrization

The model was run in non-stationary mode at mean tide water level (i.e., +2.25 m compared to depth shown in Figure 1). The contribution of wave generation by wind was assessed when building the model, confirming that the computational domain was small enough to neglect local generation (not shown here). Additional tests also confirmed that triad interactions had little effect on model results off a very shallow areas near the breaking zone, which were not targeted in this study. However, depth-induced breaking was considered, using the bore-based model of Battjes and Janssen (1978) [13] extended by Eldeberky and Battjes (1995) [14]. However, given that the surf zone was not properly resolved by the 100-m grid resolution in the coastal area, here, the depth-related sink term was essentially used to avoid non-realistic wave heights onshore.

The resolved angular sector was limited from 240° to 0° (0° meaning waves coming from the north) to spare central processing unit (CPU) time. Therefore, waves propagating from other directions were neglected. This sector was chosen by analyzing measured wave data collected 6 km off Biarritz [15], which confirmed that other angular sectors' contributions were negligible (not shown). The resolved angular sector was meshed with 40 bins, thus providing a directional resolution of 3°. The simulated frequency range extended from 0.0373 Hz to 0.25 Hz and was discretized with 20 frequencies distributed such that $\Delta f = 0.1\ f$. Stationary case tests conducted with refined spectral discretization showed no noticeable difference with the above-mentioned frequency–direction discretization, suggesting that it was fine enough to properly solve the main processes involved in the studied coastal wave propagation. The computational time step used was 15 min while the output time step was set to 3 h. The maximum number of iterations in the SWAN internal sweeping procedure was 50.

## 2.5. Validation

Figure 2a provides a scatter plot comparing simulated significant wave heights (Hs) and the five-year record of the Bayonne wave buoy (at 20-m depth). For this sample, the simulation tended to slightly overestimate the significant wave height between 0 and 3 m. However, the coefficient of determination computed (0.87) showed a strong correlation between measured and simulated data. Both the normalized bias indicator (noted NBI) and the symmetrically normalized root-mean-square error introduced by Hanna and Heinold in 1985 (noted HH) [16,17] were low (0.09 for NBI and 0.20 for HH). This confirmed that wave height was reasonably well simulated in our hindcast at the buoy location.

The scatter plot in Figure 2b represents the simulated energy period $T_e$ versus the energy period obtained from measurements. From this illustration, the model seemed to overestimate the energy period. Even if the coefficient of determination was lower than for Hs (0.72), the model fairly estimated the energy period between 10 and 14 s (i.e., where the density of point was the highest). The bias indicator NBI was 0.12. The error HH was, however, higher compared to the Hs (0.27). This is rather consistent with Lewis and Cahill, who already highlighted that the wave period is a parameter harder to estimate compared to the wave height [18].

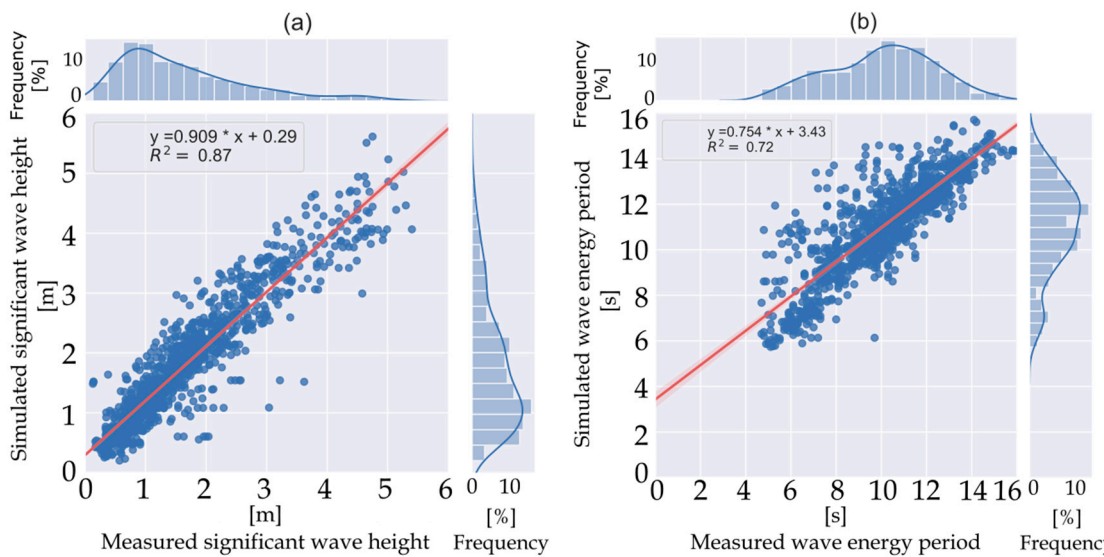

**Figure 2.** (**a**) $H_{m0}$ scatter plot; (**b**) $T_e$ scatter plot.

Simulated energy flux was calculated by integrating energy flux density over frequency and directional spaces, i.e., $P_w = \frac{\rho\,g}{1000}\int_0^{2\pi}\int_0^\infty C_g(f,D)E(f,\theta)\,df\,d\theta$. The measured energy flux was approximated by the following expression based on spectral bulk variables: $P_w = \frac{\rho g^2 H_{m0}^2 T_e}{64\pi}$, where $T_e = T_{m-1;0}$ is the energetic period. This formulation corresponds to a deep-water configuration, which may not be very accurate in the buoy location, but which had to be used due to the lack of detailed spectral information in the field data. This may have contributed to the obtained simulation/measurements energy flux differences. Moreover, the energy flux discrepancies (HH value of 0.39) were more important than for Hs because $P_w$ depends on the square of the wave height; thus, the error in $H_s$ was also squared when considering the error in $P_w$.

## 3. Results

### 3.1. Spatial Distribution of the Coastal Wave Energy Resource

The new hindcast simulation performed in this study provides a refined description of wave power spatial distribution across the south Aquitaine shelf. This detailed information is of great interest for WEC future development.

Figure 3 shows the spatial distribution of the mean wave power in the studied domain, obtained from the average over the 44 years of the hindcast simulation. This plot reveals a remarkable difference in the wave energy resource between the north ($y > 160,000$ m) of the domain and the southern part, with significantly higher energy in the north. The delimitation between both areas corresponds to the Capbreton canyon, which cuts the domain in two following the $x$-direction around $y$ ~160,000 m. North of the canyon, the mean wave energy flux ranged from 25 to 34 kW/m with an average of 27 kW/m. South of the canyon, the wave energy flux progressively decreased to values between 10 and 25 kW/m with an average around 20 kW/m. This decrease was mainly due to refraction of the wave field over the strong bathymetric gradients on each side of the canyon, which tended to focus energy in the north while creating a "shadow area" in the south. It can be noted here that the refined grid resolution over the specific bathymetric feature of the canyon seemed to be crucial to capture the related impact on the local wave resource, whereas this effect could be underestimated or even ignored by coarser regional simulations. Another contribution to the north–south coastal energy gradient may have arisen from the protective effect of the Spanish coast for wave systems propagating from the southern part of the Atlantic basin with west incident direction. However, these systems are not dominant in the studied region, suggesting a more limited impact on the obtained spatial distribution.

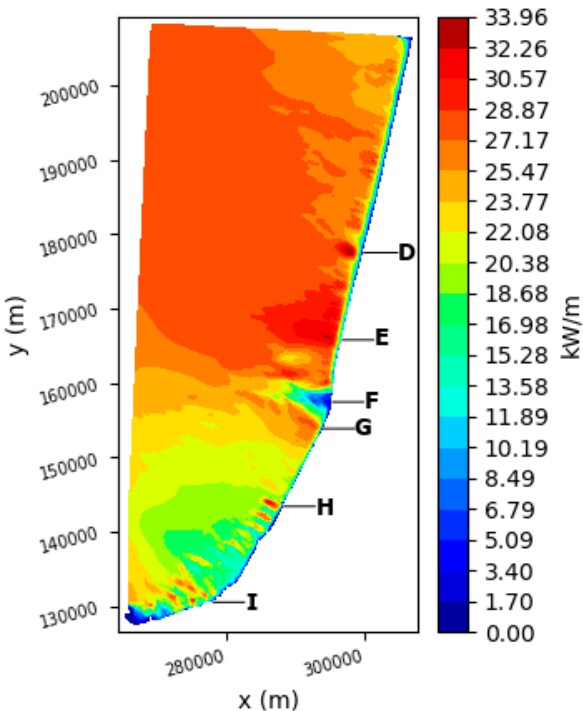

**Figure 3.** Spatial distribution of wave energy flux averaged (kW/m) over 1958–2001.

Upon examining the results further onshore in Figure 3, the effect of wave–bottom interactions was found to further intensify, resulting in a very heterogeneous wave energy distribution. Along the coast, some areas clearly focused wave energy while others received significantly less energy, mostly due to local refraction by the bathymetry. For instance, due to refraction over the Capbreton submarine canyon, wave power was very weak (5–6 kW/m) in front of Capbreton (see the location of the Capbreton Marina, noted F in Figure 3). Conversely, right in the south and in the north of Capbreton, energy was focused and power reached 25–27 kW/m in the Capbreton/Labenne area in the south (noted G in Figure 3) and more than 30 kW/m in front of Hossegor/Seignosse zone (noted E in Figure 3), with the latter being famous for its world-class surfing spots. Another hotspot of wave power was found in front of Vieux-Boucau approximately 20 km north of Capbreton (noted D in Figure 3). In this place, a maximum mean wave power of 34 kW/m was obtained, which was the maximum over the whole studied domain. In the southern half of the domain ($y < 160,000$ m), very localized spots with higher wave energy were obtained, here again due to refraction over the complex bathymetry in this mixed sandy/rocky region. Such wave energy "patches" were found, e.g., along Anglet shore close to the Adour estuary (noted H in Figure 3), or in the Saint Jean de Luz nearshore area (noted I in Figure 3), which is also known for a localized bathymetry focusing effect (e.g., the so-called Belharra giant wave).

To further quantify these strong local variations in wave energy, Figure 4 depicts the longshore profiles of wave energy resource following 20-m (top) and 10-m (bottom) isobaths. The key points mentioned above are clearly visible again on this plot, especially the remarkable drop in wave energy related to the Capbreton canyon. Furthermore, the wave energy maximum located off Vieux-Boucau can be seen at 10-m depth. Another interesting feature when comparing results at 20-m and 10-m depths was the shoreward increase wave energy at this point, resulting in the 34 kW/m maximum at 10-m water depth. This emphasizes again the interesting additional information provided at the local scale by the refined computational grid. Moreover, it suggests that shallow-water wave focusing may have generated local maxima in wave energy that exceeded the offshore resource, confirming that the maximum resource was not always to be found offshore.

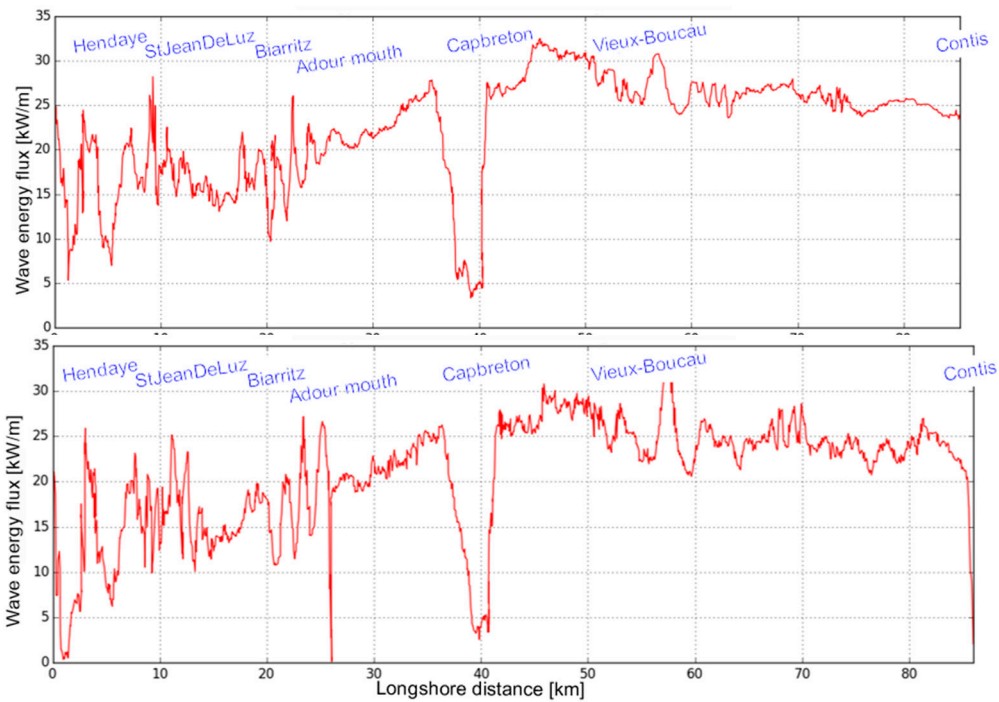

**Figure 4.** Mean energy flux longshore profiles following isobaths at 20 m (top) and 10 m (bottom).

## 3.2. Offshore and Nearshore Wave Parameter Distribution

In view of the results of the previous section, two hotspots are specifically studied in this section: (i) Point A in Figure 1, which benefits from optimum incoming wave energy in the offshore part of the domain; (ii) the Hossegor/Seignosse area, which appears to be the main nearshore energy hotpot. Firstly, Figure 5 shows the distribution of the annual wave energy depending on the significant wave height Hm0 and the energy mean period Te. The wave energy was obtained by integrating the wave power over the time frames corresponding to the corresponding wave parameters. Figure 5 shows that most of the annual energy was provided by sea states with wave heights ranging from 2 to 5 m and wave periods from 10 to 15 s at point A. The distribution was a little bit different at the Hossegor/Seignosse hotspot with larger heights and period ranges involved, i.e., 2–10 m and 8–18 s, respectively, although most of the energy was still carried by waves with characteristics similar to point A. At both hotspots considered, coarsely, half of the annual energy was roughly provided by numerous sea states with moderate energy fluxes ($Pw < 50$ kW/m), while the other half was generated by rarer events of larger intensities. Remarkably, the latter were more frequent in the Hossegor/Seignosse area than at the offshore location considered. Thus, the coastal propagation, especially the focusing on Hossegor/Seignosse, tended to shift the availability of the wave resource toward larger wave heights and periods, in addition to an overall broadening of range of wave parameters with significant energy flux compared to the offshore.

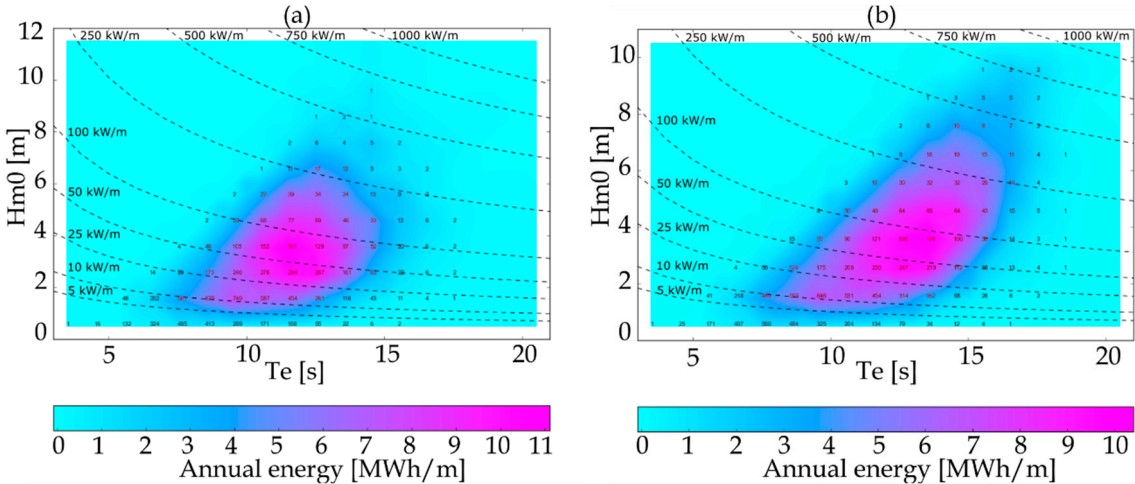

**Figure 5.** Annual wave energy in MWh·m⁻¹ versus wave height and energy period: (**a**) point A; (**b**) Hossegor/Seignosse. Dashed lines represent isolines of energy flux.

The annual energy distribution is also plotted in terms of significant wave height and peak direction in Figure 6. At point A, the energy was provided by a directional window between 280° and 310°, with most of the energy being provided by a 10° angular sector between 290° and 300°. At Hossegor/Seignosse, the energy was provided by a directional window between 275° and 305°, while most of the energy was provided by a 10° angular sector between 285° and 295°. Here again, this new set of information about the local wave resource may be of interest for WEC design and sizing, as well as efficiency evaluation at a specific installation point in the coastal area.

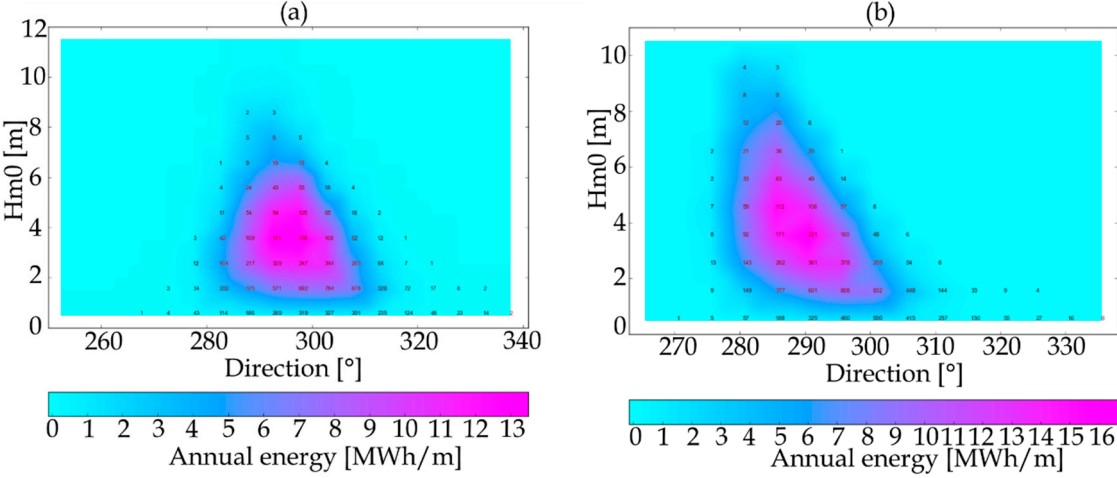

**Figure 6.** Annual wave energy in MWh·m⁻¹ versus wave height and direction: (**a**) point A; (**b**) Hossegor/Seignosse.

### 3.3. Time Variability of the Coastal Wave Energy Resource

We now examine the time variability of the resource. By considering seasonal variations, Figure 7 firstly shows the distribution of wave power in winter and summer. In winter, the repartition of hotspots was similar to that obtained when considering the entire year. A maximum winter wave power of about 62 kW/m was obtained in two locations. Although the wave resource was of the same magnitude, the spatial extent of the hotspots was different. The hotspot off Vieux-Boucau in the north was much smaller than the hotspots off Capbreton/Hossegor. In the southern part of the domain, the wave power ranged from 20 to 40 kW/m outside very local focusing points.

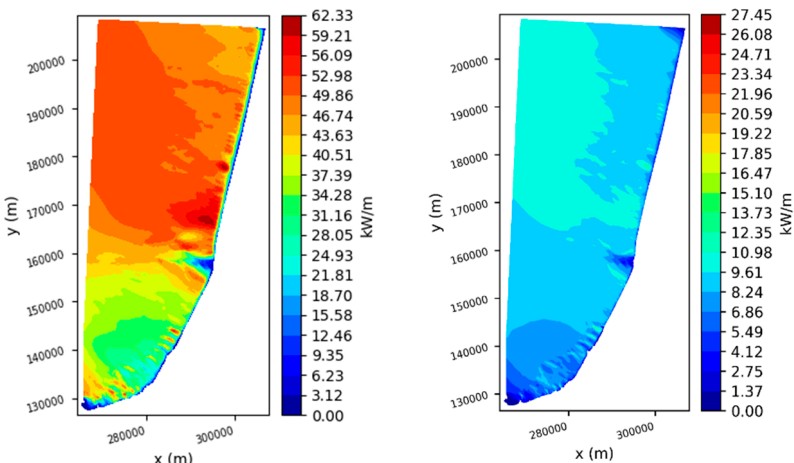

**Figure 7.** Spatial distribution of wave energy flux (kW/m) in winter (left panel) and summer (right panel) averaged over 1958–2001.

In summer, the situation drastically changed, with very low values of wave energy. The effect of the submarine canyon on the distribution of the wave energy resource is still visible in the right panel of Figure 7; however, there were no more areas with a strong focusing effect, and the discrepancies between the northern and southern part of the domain were also not as important as for the winter season. North of the submarine canyon, the wave power varied between 8 and 11 kW/m, while, in the south, the values were quite similar, ranging from 5 to 10 kW/m. Comparing these values to the mean values in winter, an energy decay of about 80% was observed in the north and an energy decay of roughly 75% was observed in the south.

The computation of the variability index introduced by Cornet [19] was performed to generalize the analysis of the variability to the whole domain. The coefficient of variation (COV) was obtained by dividing the standard deviation of the monthly mean wave power by its mean value over 44 years. Therefore, a COV equal to 1 corresponds to the case where the standard deviation is equal to the mean value. Examining the COV map plotted in Figure 8, we can observe that the temporal variability of the wave energy flux was rather uniform over the whole domain. However, it can be noted that the variability was slightly higher north of the Capbreton submarine canyon.

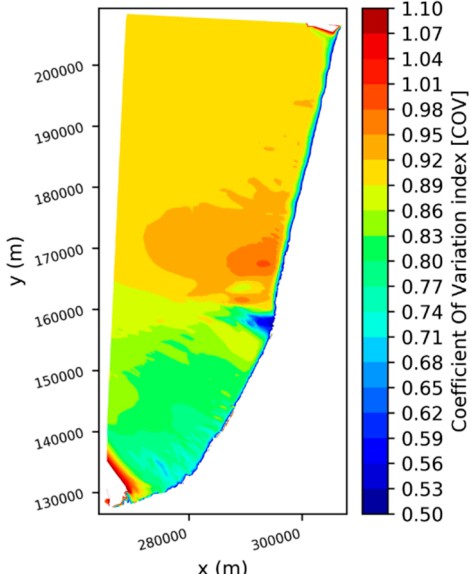

**Figure 8.** Spatial distribution of wave energy flux temporal variability represented by the coefficient of variation COV.

The same analysis was carried out for the seasonal variation index (SV) and the monthly variation index (MV). Here, considering the 44-year mean wave energy flux, the seasonal variability index was obtained by computing the difference between the mean values of the most energetic season (December–February) and the mean values of the least energetic one (June–August), normalized with respect to the annual mean. For the monthly variation index, the same computation was done using the most and least energetic months.

Figure 9 reveals the same pattern for the spatial distribution of the wave energy variability indexes as in Figure 8, with an accentuated separation between the south and the north of the Capbreton submarine canyon. In the southern part of the domain, SV ranged between 1.1 and 1.3 and MV ranged between 1.3 and 1.7. In the northern part of the domain, SV ranged between 1.4 and 1.7 and MV ranged between 1.8 and 2. Here again, the variability appears to be higher in the northern part of the domain. Additionally, we can observe that the areas with the highest wave energy resource described in Section 3.2 of the present study were also the areas with the highest variability.

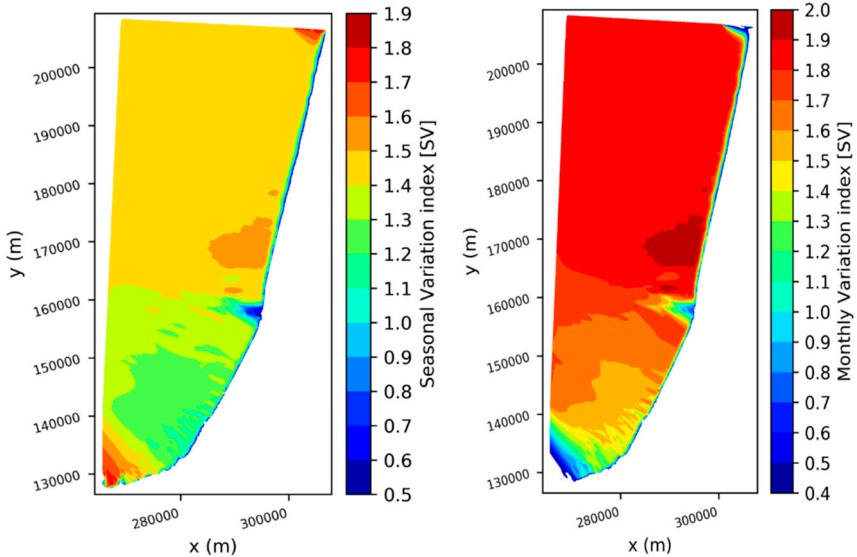

**Figure 9.** Spatial distribution of seasonal variability index of wave energy flux (SV) (**left panel**) and monthly variability index of wave energy flux (MV) (**right panel**).

Given the rather long duration of the coastal wave hindcast produced for this study, we examined the temporal variability in the wave resource to identify if a long-term trend was present. In fact, a WEC device has a lifetime of some decades; therefore, its design must fit the local resource to perform optimally over the whole period of exploitation. As shown in previous paragraphs, the south Aquitaine coastal shelf can be divided into two areas with different wave energy levels, namely, north and south of Capbreton canyon. To assess the long-term variability of the resource in these two areas, two points were specifically studied, one in the north (located offshore of Capbreton at 23.9-m water depth, represented by green squares in Figure 10) and the other one in the south (located offshore Bayonne breakwater at 26.4-m water depth, represented by red triangles in Figure 10). In Figure 10, extractions of the annual wave energy flux, inter-annual variability between two consecutive years, annual wave energy maximum, and the 98[th] percentile of the annual wave energy on these two control points are plotted for the 44-year hindcast duration. Linear regressions are also plotted to assess the evolution of these parameters over the hindcast duration.

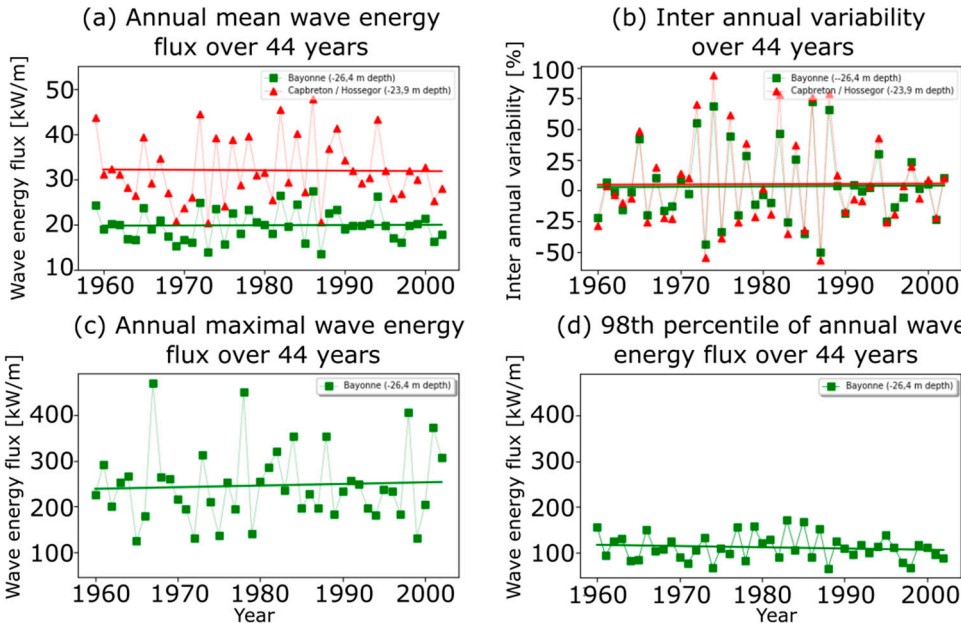

**Figure 10.** (**a**) Annual wave energy flux for the Bayonne (green squares) and Capbreton (red triangles) locations over the 44-year hindcast period. (**b**) Inter-annual variability between two consecutive years as a percentage. (**c**) Annual maximal wave energy flux for the Bayonne location. (**d**) The 98th percentile of the annual wave energy flux for the Bayonne location.

As expected, the levels of wave energy observed in Figure 10a were different at the two considered points. At the Bayonne location, wave power ranged between 15 and 25 kW/m with an average value around 20 kW/m. At the Capbreton/Hossegor location, the wave power ranged between 20 and 47 kW/m with an average value around 30 kW/m. Between two consecutive years, the resource was shown to vary significantly. The figure also highlights the fact that, even if the two points present different seafloor characteristics (sandy seafloor with small bathymetric gradients in the north and rocky seafloor with sharp bathymetric gradients in the south), the wave power evolved in a similar way, meaning that the inter-annual variability depended highly on the forcing. The similitude in the behavior of the wave energy was confirmed by Figure 10b, where the inter annual variability is plotted for the two locations. Although the evolution was almost identical, it appears that the inter annual variability was slightly higher at the Capbreton/Hossegor point. The relative variations in Capbreton/Hossegor ranged between −57% and +93%, while, in the Bayonne location, the variations ranged between −50% and +72%.

Looking for long-term trends in the plotted timeseries, Figure 10 shows that the strong inter-annual variability and the absence of repetition patterns in this variability do not allow making conjectures on long-term resource evolution. When computing linear regressions, it appears that the linear correlation coefficients are extremely low, indicating that a linear evolution of the wave energy resource is to be discarded. The slopes of these regressions, although close to 0, are not relevant given the absence of any linear behavior in the simulated timeseries. Higher-order regressions were also tested (2, 3, 4) with the same results, i.e., a very low regression coefficient.

## 4. Discussion

The extensive dataset provided by the high-resolution coastal wave hindcast allowed an assessment of the local wave energy resource variability in both time and space. When studying the spatial variability of the resource, the most remarkable feature was the strong effect of the Capbreton's submarine canyon. The latter induced a separation of the coastal area in two domains: one domain lying north of the submarine canyon where wave energy was focused, and the other one in the south, where the coast seems to be sheltered by the refraction over the submarine canyon. Furthermore, the

strong longshore variability in the southern part of the studied area was also noticeable. Compared to the northern part, the wave energy fluctuated much more over the locally very heterogeneous bathymetry. While the northern part is characterized by a sandy coast with more uniform bathymetry, the southern part is composed of a rocky seafloor related to steep bathymetric gradients. Although these gradients were smaller than those observed in the submarine canyon, they had a strong impact in terms of local focusing effect of the wave energy resource in the coast south of Bayonne. Overall, local wave refraction appeared to be the major process controlling local wave resource in the coastal area. Although a coastal region was potentially shown to have an interesting wave energy overall resource, local effects in finite depth may have strongly modulated the resource, sometimes by a factor of two. This may be an opportunity to increase the production or a threat for the system survivability, depending on the configuration. In any case, results obtained in this study strongly support the need for high-resolution coastal wave investigation to properly optimize coastal WEC installation.

Regarding long-term temporal trends, in light of the present work, it seems not possible to draw any conjecture about future wave energy resource evolution. Over the 44 years covered by the hindcast, the wave energy was shown to vary a lot between two consecutive years, with variations values reaching up to 93%. However, this very high variability could be explained by more global phenomena like the North Atlantic Oscillation (NAO). Good correlations were obtained between wave energy resource and climate indices [20]. Understanding and better quantifying the link between wave energy resource and the pressure anomalies, together with improving the ability of predicting accurately those pressure anomalies, would benefit the ability to predict future wave energy resources. Even if multiannual predictions of the NAO are not yet achieved, recent studies showed that it might be possible using enhanced models capable of better capturing the behavior of the atmosphere and its driving factors.

The present study also showed the wave conditions to be very different along the year. While, in winter, the wave energy resource was very high, in summer, it decreased significantly. This underlines the fact that annual mean values of wave energy should not only be considered when designing a WEC project as they do not carry all the information about the resource, especially its intra-annual variability. It is important to emphasize that the expected energy production of a WEC will be very different between winter and summer, and that policymakers should not expect a WEC device to produce an average amount of electricity all over the year. Moreover, these relative variations were also shown to be similar between the northern and southern parts of the domain. However, even if the relative variations were equal, the absolute variations were not, due to the different wave energy levels. Thus, the variations in the southern part of the south Aquitaine coastal shelf ranged between 2 and 12 kW/m between two consecutive years, while, in the north, these variations ranged between 2 and 27 kW/m. Therefore, from a WEC installation point of view, it seems important to consider two aspects, namely, the wave energy overall resource and the resource temporal stability. Indeed, previous studies already mentioned that stable energy levels might be more interesting than fluctuating high-energy values [21]. Nevertheless, this study highlights the interest of a long hindcast period as it can provide valuable information to WEC designers about several important aspects. These variations are interesting both for an energy production optimization process or for resistance design against extreme events (e.g., winter storms).

During the implementation of the model, some simplifying hypotheses were made. Although their contributions were shown to be negligible due to the small domain studied, various coastal phenomena were disabled (e.g. bottom friction, wave–wave interaction, white capping effect). Further refined studies on, e.g., a specific location potentially eligible for WEC installation, could need further research to include their impact on the wave energy. In addition, a strong hypothesis was made concerning the bathymetry. Due to the lack of continuous bathymetric information over the 44 years of the simulation, the bathymetry was assumed to be constant. This may be a strong hypothesis if the depth considered for the WEC deployment is very shallow (< 20 m), especially considering the sandy seafloor north of Bayonne. In such locations, sediment transport could modify the bathymetry, which,

in turn, could affect the wave propagation, thus resulting in variations in the wave energy resource [22]. Dredging works are also often conducted near the Bayonne harbor and may be important to consider. Finally, a last source of discrepancies could be related to numerical aspects, especially numerical diffusion and/or limiters used by SWAN, which may affect the representation of wave refraction over steep bathymetric gradients [12,21].

## 5. Conclusions

In this paper, a local wave energy assessment was provided in the south Aquitaine coastal area. Using a high-resolution unstructured spectral wave model implementation, a 44-year coastal wave hindcast was built. This hindcast was validated against wave-buoy data collected over five years. The spatial mapping of the resource underlines the great impact of the submarine canyon of Capbreton, resulting in a remarkable south–north energy gradient in the region; the average power was 27 kW/m north of the canyon and 20 kW/m in the south (i.e., a variation of about 35%). Similarly, the presence of such bathymetric gradients induced specific wave energy focalization in hotspots, with the most important being found off the coast of Vieux-Boucau and between Hossegor and Seignosse. Alongside the study of the spatial variation of the wave energy resource, the quantification of temporal variability was addressed, benefiting from the rather long duration of the hindcast dataset. The variability of the resource was shown to be very high at both intra-annual (decay of 80% between summer and winter) and inter-annual levels (up to 93% variation between two consecutive years). However, no long-term trend was identified due to chaotic inter-annual variability.

An extra step toward the installation of WEC devices in the French Basque coastal area would require precisely defining their locations. The energy-focusing localizations described in this paper would, therefore, need higher-space-resolution meshings to capture very local variations, which could be part of further investigations.

**Author Contributions:** Conceptualization, S.A., M.D., and P.L.; methodology, S.A., M.D., and P.L.; software, P.M. and X.L.; validation, X.L., P.M., S.A., and M.D.; formal analysis, X.L., S.A., M.D., and V.R.; investigation, X.L. and S.A.; writing—original draft preparation, S.A.; writing—review and editing, X.L., S.A., M.D., and V.R.; visualization, P.M. and X.L.; supervision, S.A., M.D., J.M., and V.R.; project administration, S.A., M.D., and J.M.; funding acquisition, S.A., M.D., and J.M. All authors read and agreed to the published version of the manuscript.

**Funding:** The authors acknowledge funding of this work by the Aquitaine Euskadi Region through the WAKE project (Steps toward joint capacities for wave energy exploitation in Aquitaine Euskadi). Volker Roeber acknowledges financial support from the Isite program Energy Environment Solutions (E2S), the Communauté d'Agglomération Pays Basque (CAPB), and the Communauté Région Nouvelle Aquitaine (CRNA) for the chair position HPC-Waves.

**Conflicts of Interest:** The authors declare no conflicts of interest.

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
