# Peer review of "Wave Energy Assessment in the South Aquitaine Nearshore Zone from a 44-Year Hindcast"

_jmse, doi:10.3390/jmse8030199_

Round 1

Reviewer 1 Report

The authors have made a considerable effort to address all the issues in my original report.  I find the responses and implementations acceptable, and am pleased to recommend acceptance of the manuscript for publication.  The English may require some further polishing - I will any copyediting to the editor's judgement.

Author Response

We thank the reviewer for its positive feedback on our manuscript.

Reviewer 2 Report

Following most of the reviewers' suggestions, the authors have improved the
manuscript. However, I suggest to finalize it according to the following minor comments.

Figure 1: Points B and C reported in the figure never appear in the text. I suggest to eliminate the label from the figure or to explain their role into the text and in the figure caption. Are they 'the most onshore BOBWA output points as illustrated in Figure 1'?, line 74. In the following line you refer to 10 BOBWA points and not to three points.

line 96: Define the acronym BSBT

line 145: 'computed' should be changed with 'measured' or 'observed'

Figure 2: add y label (% ?) in the lateral histograms

Reviewer 3 Report

The authors have taken into account all my comments and I think that the paper is suitable for publication in JMSE

Author Response

We thank the reviewer for its positive feedbacks on our manuscript.

This manuscript is a resubmission of an earlier submission. The following is a list of the peer review reports and author responses from that submission.

Round 1

Reviewer 1 Report

See attached file

Reviewer 2 Report

In the present paper the authors evaluate wave resource availability in the southern part of Biscay gulf.
The data used to perform this evaluation are produced through SWAN simulations carried out over a high-resolution
unstructured grid in order to highlight the effect of small scale bathymetric features.
The work is interesting but I suggest some minor revisions.

line 60: could be useful to specify the resolution of WWIII data used for SWAN boundary conditions
line 111-114: "Wave generation by wind is neglected by considering that the computational domain is relatively small. For the same reason, white-capping effect, bottom friction and quadruplets interactions are also turned off in the model. Triad interactions are also neglected" --> necessary to add citations
line 128-130: it would be useful to explain more in depth how these tests are performed. Since I am not a statistician, I find difficult to understand what the indexes are telling, and what is the hypothesis they are referring to.
line 324-326: "Although they were expected to be negligible due to the small domain studied, various coastal phenomena have been disabled such as bottom friction, wave-wave interaction, white capping effect, local wave generation by wind" --> necessary to add citations
line 320-331: "This is a strong hypothesis especially considering the sandy seafloor north of Bayonne. In such locations, sediment transport could modify the bathymetry which, in turn, could affect the wave propagation thus resulting in variations in the wave energy resource" --> necessary to add citations

To improve the understanding of the paper I suggest enlarging text presents in Figures 4, 5 and 6 and to mark in a map the places cited in the text (i.e. Hossengor/Seignosse etc.) instead using x/y coordinates.

Reviewer 3 Report

Dear Editor,
I read carefully the manuscript entitled "Wave Energy Assessment in the South Aquitaine Nearshore Zone from a 44 Year Hindcast" submitted for publication on the Journal of Marine Science and Engineering. The paper deals with wave energy assessment and characterization along the south Aquitaine coast. The subject of the paper is of interest for JMSE even if this topic right now it is not really on the edge of the actual research of renewable wave energy, but it would be useful for practical applications. The paper is reasonably structured and well written. Anyway I think that it does not deserve publication in the present form mainly because the data analysis is a bit poor if we take into account the extension in space and time of the wave hindcast developed by the authors. After a re-organization and different integration I think that the manuscript could be taken into account for a new re-submission. I provide a list of my comment/concern here below.

- I would like to have more info about the numerical model implementation: which kind of numerical method has been employed? which time step has been used? The authors made any sensitivity analysis about the reliability of the results?
- I agree with the authors that some simplification should be done in order to spend as little time as possible for computational effort, but I think that may be the effect of triads and local wind generation should be investigated. Could the author provide some test cases where strong winds and mixed seas are taken into account in order to check that the effect of triads and local wind generation are negligible?
- Figure 2: please add the label of the quantity directly on the graph, use bigger fonts, use color scale for density of data in the scatter plot. the plot about the energy flux is useless because it is based on the previous two, so please omit it
- Together with graphical representation of the accuracy of the results (figure 2) please provide quantitative evaluation through the estimate of typical statistical error indicators such as Normalized Bias NBI, Correlation Coefficient, symmetrically normalized root mean square error HH introduced by Hanna and Heinold (1985)
- Figure 4 should be better done (bigger character and quality). Add label in the figure directly. The same comment applies for Figure 5 and 6
- Because some comment have been done about variability of the resource I would expect the calculus of COV, MV and SV, as introduced by Cornett et al (2008) in the whole the domain
- I do not agree that it is not possible to check for trend in the wave energy resource as stated by the authors and I would like to have some estimate of possible trends of different quantities (annual mean, annual maxima, annual percentiles such ad 98%)